# Youth Bullying and Suicide: Risk and Protective Factor Profiles for Bullies, Victims, Bully-Victims and the Uninvolved

**DOI:** 10.3390/ijerph19052828

**Published:** 2022-02-28

**Authors:** Ching Kwan, Clifford Wong, Zhansheng Chen, Paul S. F. Yip

**Affiliations:** 1HKJC Centre for Suicide Research and Prevention, Hong Kong 999077, China; ikwan@hku.hk (C.K.); wlhcliff@hku.hk (C.W.); 2Department of Psychology, University of Hong Kong, Hong Kong 999077, China; chenz@hku.hk

**Keywords:** suicide, adolescents, bullying, risk factors, protective factors

## Abstract

Bullying is closely associated with suicide. This study validates mixed evidence on whether young bullies, victims, bully-victims, and those uninvolved in bullying differ in suicidality, risk, protective factor profiles, and predictors of suicide. A total of 2004 Hong Kong adolescents and young adults completed the Hong Kong Online Survey on Youth Mental Health and Internet Usage in 2018. Bullies, victims, and bully victims, as opposed to the uninvolved, were found to possess higher tendencies of suicidal thoughts and behaviors. They had more distinct rather than overlapping risk and protective factor profiles yet shared psychological distress and diagnosis of a psychiatric disorder as common predictors of suicide. The results indicate that suicide screening assessments and training to detect common suicide predictors can benefit youngsters regardless of their bullying involvement. From the discussion, group-specific interventions include restorative justice approaches to promote reintegration and help-seeking among bullies, peer, and professional support programs geared towards lowering victim isolation and equipping gatekeepers such as teachers with skills to connect with both bullies and victims.

## 1. Introduction

Suicide is a serious public health concern among adolescents and young adults, remaining the second leading cause of death among those 15 to 29 [1]. Interpersonal experience is an important contributing factor to suicide [2,3]. For instance, experiences of being ignored and/or excluded can trigger suicidal thoughts [4]. Thus, it is not surprising that bullying, unwanted aggressive behavior that inflicts distress or harm on targeted individuals, is closely associated with suicide [5]. Substantial evidence has indicated that any bullying involvement heightens the risks of suicidal ideation, suicidal behavior, and poor mental and physical health outcomes [6,7,8].

With between 20% and 56% of youth involved in bullying annually, bullying was most severe during 11–13 years of age, with a more equal male: female ratio involved in bullying during adolescence [9,10,11]. Their involvement can be divided into those who bully others (i.e., bullies), are bullied (i.e., victims), and both bully and are bullied (i.e., bully-victims) [9]. Each of these three categories (i.e., the involved) and the uninvolved were found to be distinct from one another in a range of psychosocial variables and problem behaviors, including peer influences, attitudes towards deviance, and school-related functioning [12]. Yet, mixed evidence regarding the associations between these four groups and suicide was found. Borowsky et al. [13] concluded that suicidal ideation and attempt rates were the lowest at 6.3% and 1.2% for the uninvolved, 16.5% and 5.0% for bullies, 21.8% and 6.5% for victims, and the highest at 26.1% and 11.1% for bully-victims. In contrast, other studies found that frequent suicidal ideation and self-injurious behavior were more strongly associated with victim and bully victims than bullies [8,14], or failed to find higher levels of suicidal ideation for bully victims than those uninvolved in bullying [15]. Given the theoretical and practical significance for better understanding which group(s) involved in bullying are at higher risk for suicide, it is necessary to compare these groups concerning suicidal ideation and behavior directly.

Additionally, bullying involvement coupled with risk factors increase youngsters’ chances of suicidal behaviors [5]. However, few studies have validated the risk, and protective factors associated with different types of bullying involvement or simultaneously tested the different parties involved in bullying across a wide range of factors. Common risk factors for bullies, victims, and bully victims include high levels of depression, emotional distress, self-harm [16]. Specific risk factors include high levels of anxiety for bullies, high domestic violence for victims, weak family attachment, females with poor empathetic understanding for bully victims, and alcohol abuse among victims and bully victims [17,18]; specific protective factors include prosocial involvement for bullies, high self-esteem, perceived social support, family togetherness, high academic performance, and peer support for victims [19,20,21]. Yen, Liu [17] discovered that risk or protective factors that contribute to adolescent suicide attempts and suicide differed across bullies, victims, and bully victims, yet as only seven factors were measured, they highlighted the importance of further studies to evaluate the predictive effects of risk and protective factors.

In summary, current evidence bolsters bullying involvement’s strong association with suicide. Yet, further investigation is needed to validate whether certain or all groups involved in bullying are at higher risk for suicide, to identify the risk and protective factor profiles of different groups involved in bullying, and to determine if each group’s risk and protective factors uniquely predict suicidality.

The present study aims to address the current gaps in research. A comprehensive examination of the risk and protective factor profiles is conducted for the following groups: (1) bullies, (2) victims, (3) bully victims, and (4) the uninvolved. This study proposes the following hypotheses:Bullies, victims, and bully victims will exhibit higher tendencies of suicidal thoughts and behaviors than those uninvolved in bullying;Bullies, victims, bully victims, and those uninvolved will possess group-specific risk and protective factor profiles;Bullies, victims, bully victims, and uninvolved individuals’ distinct risk and protective factors will uniquely predict their suicidal thoughts and behaviors.

## 2. Materials and Methods

### 2.1. Participants

Participants (*N* = 2004) consisted of 32% males and 68% females with a mean age of 23.27 (SD = 5.14). The majority were students (74%) with an educational level of post-secondary or above (84%). Eligibility to participate included residence in Hong Kong, literacy in Chinese, and an age between 11 to 35. Youth age is not officially defined in Hong Kong [22]. Therefore, participants’ age range followed the definition set by OpenUp, a large-scale online text platform serving young people between 11 to 35 in Hong Kong [23]. They were categorized according to whether they have bullied or been bullied by others (including online bullying) in their lifetime into the bully (*n* = 119), victim (*n* = 274), bully victim (*n* = 274), and uninvolved (*n* = 1198) groups.

### 2.2. Measures

Demographic information was collected on gender, age, student and occupation status, and the highest educational attainment achieved. Evidence-based risk and protective factors for suicide were also measured: risk factors include exposure to stress [24], stigmatizing attitudes [25], common mental health disorders [26], social withdrawal [27], and help-seeking barriers [28]; protective factors include common help-seeking sources [29,30].

Participants responded to “in the past 4 weeks, did you experience distress in the following areas of your life?” across eight areas (academic, job, financial, social life, physical well-being, mental well-being, relationship with family, and relationship with spouse). Ratings ranged from 1 (not at all) to 5 (very serious), or N/A (not applicable) for non-relevant stressors. Stressors rated as N/A were recoded as 0 and included in the analyses to account for the overall distress level of the sample.

The Stigma of Suicide Scale-Short Form (SOSS-SF) measured respondents’ stigma towards suicide as they rated their level of agreement on descriptors of a typical person who completed suicide [2]. As a person’s attitude towards suicide accounts for variance in suicidal ideation that cannot be explained by hopelessness or symptoms of depression [31], positive attitudes toward suicide were predictive of suicide risk status [32]. Items were scored from 1 (strongly disagree) to 5 (strongly agree), with higher scores indicating higher levels of stigma. Four stigma subscales were measured: glorification, isolation, disgrace, and selfishness. The four-factor, 12-item SOSS-SF scale was employed as it was found to be a better fit than the original three-factor, 16-item scale for the Hong Kong population [33].

Respondents provided “yes” or “no” indications on whether they have bullied or been bullied by others in any way (including online bullying), considered suicide, attempted suicide, and performed deliberate self-harm by injuring themselves intentionally in their lifetimes. Those who have considered suicide then completed the Suicidal Ideation Attributes Scale (SIDAS).

SIDAS measured respondents’ severity of suicidal thoughts in the past month [34]. Participants rated the frequency of suicidal thoughts, their control over their thoughts, how close they were to a suicide attempt, how tormented they were about suicidal thoughts, and how much the suicidal thoughts interfered with their lives on a 5-point scale (1 = never, 5 = very). Higher total scores reflected greater severities of suicidal ideation. The scale was found to have a high level of internal consistency (Cronbach’s α = 0.91).

The 12-Item Chinese Health Questionnaire (CHQ-12) identifies the degree of psychological distress among respondents [35,36]. Items were scored on a 4-point scale from 1 (not at all) to 4 (much more than usual), and higher total scores indicate greater psychological distress [37]. The scale was found to have a high level of internal consistency (Cronbach’s α = 0.85).

Social withdrawal was measured using the Social Engagement-Hikikomori Scale’s first two questions on whether participants “spend most of the day and nearly every day confined at home” and “persistently avoid social situations and social contact” [38]. Those who selected “yes” for both questions and had not been diagnosed with social phobia, major depressive disorder, schizophrenia, or avoidant personality disorder were categorized as socially withdrawn [39].

Participants were asked if “you may have mentioned some distressing issues or life difficulties. Concerning those issues, did you seek help from any of the following in the past 4 weeks? (Check all that apply)”. Help-seeking sources included family members, friends/classmates/colleagues, lover/life partner, teachers/tutors, free hotline support, medical professionals, social workers/counselors, religious services, online friends, and online social services.

Participants who selected none of the help-seeking sources checked all applicable reasons for not seeking help, including “not sure how others will think of me”, “I don’t know where to seek help”, “I don’t want to bother others”, “I don’t think it is necessary”, “I don’t think anyone can help me”, “I am afraid it will leave a record and affect my future”, “I have no distressing issues or life difficulties” and “others (please specify)”.

### 2.3. Procedure

The collected data was from the Hong Kong Online Survey on Youth Mental Health and Internet Usage initiated by the Hong Kong Jockey Club Centre for Suicide Research and Prevention (CSRP) [19]. From 22 December 2017 to 15 July 2018, the survey was disseminated by CSRP, HKU, and NGOs (Caritas, Hong Kong Federation of Youth Groups, and The Boys’ and Girls’ Clubs Association of Hong Kong) through pages, emails, posters, and reminders. It was described to potential participants as a research study to gain insights into the young generation’s general well-being and usage of the internet and social services. Participants signed an informed consent form before completing an online survey of approximately 10 min. Participants could choose to withdraw anytime and were encouraged to seek help from the contact information of emotional support services and hotlines provided at the end of the survey should they experience distress. This study was approved by the Human Research Ethics Committee for Non-Clinical Faculties of HKU (Research Ethics Approval ID: E41709039).

### 2.4. Data Analyses

To identify the groups’ distinct risk and protective factor profiles, one-way ANOVA, MANOVA, Chi-squared test, and Fisher’s exact test were used to test for group differences. Fisher’s exact test was also used to account for several responses’ low frequencies, addressing issues of small sample sizes, which the chi-square test’s approximation method cannot accurately analyze [40,41]. Following significant results in omnibus tests, post hoc tests were conducted through pairwise comparisons using Tukey’s HSD correction. All tests were concluded as statistically significant if *p* < 0.05. Phi (φ) and eta-squared were computed to calculate the effect size of the results.

To examine whether each group’s distinct risk and protective factors explain significant variance in their suicidal thoughts and behaviors, hierarchical multiple linear and logistic regressions with backward stepwise AIC variable selection were performed for all groups. The *p*-level for all comparisons was set at <0.05, and all variables inputted into the four-step regression model are summarized below.

Step 1 (demographic variables): “age + gender (female versus male) + education level (above secondary school versus secondary or below) + employment status (full-time versus part-time/no work) + family structure (non-nuclear family versus nuclear family) + living with whom (living with others versus living alone)”.

Step 2 (psychological and social factors): “academic/work stress + financial stress + social life stress + physical health stress + mental health stress + family/partner stress + CHQ-12 + diagnosed with a psychiatric disorder (depression schizophrenia/social phobia + avoidant personality disorder) + social withdrawal”.

Step 3 (help-seeking behaviors): “seeking help from intimate others (family/friends/partner) + seeking help from professionals (teacher/medical professionals/social workers/religious services) + seeking help from virtual platforms (hotline/online friends/online social services)”.

Step 4 (stigma of suicide): “glorification + isolation + disgrace + selfishness”.

For logistic regressions, a quasi-complete or complete separation of data points were detected for a few variables in the bully and uninvolved group. The outcome variable separates the predictor variables to a certain degree or perfectly, resulting in large or infinite coefficients for predictor variables and their parameter estimates. To avoid biased estimates for other predictor variables, predictor variable outcomes with complete or quasi separation were included but not interpreted in the model, and some variables were collapsed into umbrella categories to reduce this issue [42].

## 3. Results

### 3.1. Socio-Demographic Profile

Table 1 provides the full results of comparisons between the bully, victim, bully-victim, and uninvolved groups. Participants (*N* = 2004) were categorized into 119 bullies (M age = 22.97, SD = 4.63), 413 victims (M age = 23.63, SD = 5.19), 274 bully-victims (M age = 23.42, SD = 5.16), and 1198 uninvolved (M age = 23.14, SD = 5.17).

### 3.2. Suicidal Thoughts and Behaviors of the Bully, Victim, Bully Victim, and Uninvolved Groups

Overall, the results support hypothesis one as bullies, victims, and bully victims demonstrated higher tendencies of suicidal thoughts and behaviors than those uninvolved in bullying. The effect of different groupings on suicidal ideation (SIDAS) was significant, F = 34.46, *p* < 0.001, partial η^2^ = 0.05. Victims (M = 15.27, SD = 8.16), bully-victims (M = 14.68, SD = 7.14) and bullies (M = 13.09, SD = 6.21) all scored significantly higher than the uninvolved group (M = 12.05, SD = 5.03) (Table 1). Significant relations were also found between participant groupings and the percentage of participants that considered suicide (*p* < 0.001), attempted suicide (*p* < 0.001), and injured themselves intentionally (*p* < 0.001). The uninvolved were significantly less likely to consider or attempt suicide than other groups, whereas the highest proportion of bully-victims injured themselves intentionally (47.08%), followed by the victim group at 38.74% (Table 1).

### 3.3. Risk and Protective Factor Profiles of the Bully, Victim, Bully Victim, and Uninvolved Groups

Support was also found for hypothesis two as the four groups displayed more distinct than overlapping risk and protective factor profiles.

#### 3.3.1. Source of Stress

The groups reported significantly different mean stress scores for seven of the eight sources of stress, including academic (*p* < 0.01), job (*p* < 0.001), social life (*p* < 0.001), physical wellbeing (*p* < 0.01), mental wellbeing (*p* < 0.001), relations with family (*p* < 0.001), and relations with partner (*p* < 0.05). Post-hoc tests revealed that the victim group’s stress scores were significantly higher than those of the uninvolved group for all significant sources of stress except for relations with a partner (Table 1).

#### 3.3.2. Stigma of Suicide

The four groups assigned significantly different scores for three of the four subscales: glorification (F(3, 2000) = 9.53, *p* < 0.001, partial η^2^ = 0.01), isolation (F(3, 2000) = 5.02, *p* < 0.01, partial η^2^ = 0.01), and selfishness (F(3, 2000) = 5.87, *p* < 0.001, partial η^2^ = 0.01) (Table 1).

#### 3.3.3. Severity of Suicidal Ideation

The effect of different groupings on suicidal ideation (SIDAS) was significant, F = 34.46, *p* < 0.001, partial η^2^ = 0.05. Victims (M = 15.27, SD = 8.16), bully victims (M = 14.68, SD = 7.14) and bullies (M = 13.09, SD = 6.21) all scored significantly higher than the uninvolved group (M = 12.05, SD = 5.03) (Table 1).

#### 3.3.4. Psychological Distress

Similarly, the four groups demonstrated varying levels of psychological distress (CHQ-12 score), F = 22.51, *p* < 0.001, partial η^2^ = 0.03. From the Tukey post hoc test, victims (M = 23.39, SD = 6.46), bully victims (M = 22.87, SD = 6.48), and bullies (M = 22.76, SD = 6.33) displayed significantly more psychological distress than the uninvolved (M = 20.79, SD = 6.18) (Table 1).

#### 3.3.5. Risk Behaviors

Significant relations were found between participant groupings and the percentage of participants that considered suicide (*p* < 0.001), attempted suicide (*p* < 0.001), and injured themselves intentionally (*p* < 0.001). The uninvolved were significantly less likely to consider or attempt suicide than other groups, whereas the highest proportion of bully victims injured themselves intentionally (47.08%), followed by the victim group at 38.74% (Table 1).

#### 3.3.6. Social Withdrawal and Psychiatric Disorders

Groups exhibited similar levels of social withdrawal. Victims showed the highest prevalence of major depressive disorder (13.08%) and schizophrenia (3.15%); bully victims demonstrated the highest prevalence of social phobia (6.20%); bullies displayed the highest prevalence of avoidant personality disorder (4.20%) (Table 1).

#### 3.3.7. Help-Seeking Sources

Apart from family members, free hotline support, and religious services, the four groups reported significantly different levels of help-seeking for the remaining eight sources. All four groups exhibited the highest percentage of help-seeking from friends/classmates/colleagues, comprising over half of each group from 50.42% (bullies) to 63.87% (bully victims). For six of the eight significant sources, victims and bully victims had higher help-seeking percentages than bullies and the uninvolved (Table 1).

#### 3.3.8. Help-Seeking Barriers

The relation between groupings and help-seeking barriers was significant for two of the eight barriers. For “I don’t think it is necessary”, (*p* < 0.001), bullies had the highest frequency of 20.17%, followed by those uninvolved at 18.70%. The uninvolved had a frequency of 17.95% for “I have no distressing issues/life difficulties”, which was significantly higher than bullies (7.56%), victims (8.72%), and bully victims (8.76%) (Table 1).

### 3.4. Predicting Suicidality from Demographics, Psychological and Social Factors, Help-Seeking Behaviors, and Stigma towards Suicide

Regarding hypothesis 3, Table 2 summarizes the mean and standard deviation of SIDAS scores, as well as the proportion of those who attempted suicide or intentional self-harm within the bully, victim, bully victim, involved (all bullies, victims, and bully victims), and uninvolved groups.

#### 3.4.1. Predicting SIDAS Scores

Table 3’s hierarchical multiple linear regression revealed that the model was statistically significant (*p* < 0.05) for all groups across the four stages, except for stage one for the bullies and stage three for the uninvolved. The goodness of fit also improves in each stage as the AIC decreases, and the positive *R*^2^ and Δ*R*^2^ suggest that the addition of variables in each stage accounts for more variation in SIDAS scores. For all groups, the highest Δ*R*^2^ were found in stage two of the model with CHQ score and diagnosis of a psychiatric disorder contributing significantly to the model (*p* < 0.05). Significant predictors between the five groups largely overlapped. Between the involved and uninvolved groups, obtaining a post-secondary level of study and viewing suicide as disgraceful significantly decreased suicidal ideation; an increase in CHQ score, diagnosis of psychiatric disorders, seeking virtual help, and glorifying suicide significantly heightened suicidal ideation.

#### 3.4.2. Predicting Suicide Attempt

Table 4 provides hierarchical multiple logistic regression summaries of factors predicting suicide attempts. The model was statistically significant across the four stages for all groups apart from stages one and four for bullies, stage three for victims, and stage four for bully victims. Except for stage four for bullies and bully victims, the increase in Δ*R*^2^ reflects that introducing variables to each stage explained additional variance in suicide attempts.

#### 3.4.3. Predicting Self-Harm

Regarding predictions for intentional self-harm, Table 5 reports findings of the hierarchical multiple logistic regression. Statistical significance for the model was found apart from stage one for the uninvolved, stage three for bullies, bully victims and the uninvolved, and stage four for bully victims. Increases in *R*^2^ and Δ*R*^2^ were found apart from stage three for bullies, bully victims, and the uninvolved, and stage four for bully victims. Similar to predictors for suicidal ideation, increases in CHQ score and diagnosis of psychiatric disorders were significantly associated with a higher prevalence of suicide attempts and intentional self-harm, and glorification of suicide was also associated with intentional self-harm for both the involved and uninvolved groups.

## 4. Discussion

The study’s results have significant implications for efforts to understand the commonalities and distinguishing factors between bullies, victims, bully victims, and those uninvolved in bullying. Support was found for Espelage and Holt [6], Kim and Leventhal [7], and Winsper, Lereya’s [8] findings on the association of increased suicidal risk behaviors with any type of bully involvement. Regarding current literature’s mixed findings on the corresponding degree of suicidality for different types of bullying involvement, victims scored significantly higher for suicide ideation, but there was no evidence that bullies, victims, and bully victims differed significantly from one another in frequencies of suicidal considerations and attempts. Contrary to our expectations, the most notable difference to emerge was that the rate of suicidal thoughts and attempts for all involved groups was approximately three times higher than statistics reported in the Borowsky, Taliaferro [11] survey. Though the formerly reported lifetime prevalence and the latter reported prevalence over the past year, the cross-national lifetime prevalence of considering and attempting suicide were only 9.2 and 2.7% [43] as opposed to 34.72% and 3.67% for the uninvolved, and averages of 58.60% and 12.74% for those involved in bullying. With all types of bullying involvement exhibiting alarmingly high rates, the evidence further justifies the need to identify the risk and protective factor profiles, as well as the predictors of suicide for these equally vulnerable groups.

Overall, the groups exhibited more distinct than overlapping risk and protective factor profiles. Group-specific differences were found for sources of stress, the stigma of suicide, diagnosis of psychiatric disorders, and CHQ scores. Similarities across the groups were found for levels of social withdrawal and most help-seeking barriers. Such profiling is especially prevalent for victims, who assigned the highest stress scores to job, social life, physical well-being, mental well-being, and relations with family. The stress hormones of adolescents increase when bullied, and chronic stress alters their brain structure with effects of increased anxiety and irregular emotional responses, as well as enhanced amygdala activity [44,45]. School bullying victims are more likely to experience higher post-traumatic stress, a lower sense of self-worth, and a stronger belief in an external locus of control [46], suggesting that victimization may be both a stressor and a trigger that heightens victims’ stress experience across different domains. Furthermore, victims are more likely to experience isolation. They are less sociable with fewer playmates in school [47], demonstrate high levels of absenteeism in the workplace [48], and are more likely to have overprotective parents who lower a victims’ ability to explore new situations with peers [49]. Their isolating experiences can explain their mirroring attribution of people who died by suicide as being isolated (SOSS-Isolation), as well as their increased tendencies to seek help outside their social circles and primary environments.

Each group’s different preferences in help-seeking sources and barriers have practical implications regarding applying effective interventions for each group. Friends/classmates/colleagues were the leading help-seeking source across all groups, suggesting that establishing positive social connections among peers can be an effective protective factor. Encouraging peer support of befriending schemes, mediating between the bully and the victim, and active listening to provide emotional support have been found to facilitate social inclusion and the development of a caring school climate [50]. In organizational settings, co-workers can provide confidential support, for instance, the practical listeners program launched by the United Kingdom’s postal service, in which staff volunteers were trained to provide social support, suggest possible courses of action, and listen non-judgmentally [51,52]. However, research has also shown that social support from friends was insufficient to shield bullied youth from the mental health difficulties they face [53]. Victims and bully victims’ higher tendencies to seek help from teachers, counselors, medical professionals, and online communities reflect the importance of training well-informed school and medical professionals. Teachers should be aware of their influence concerning reducing bullying through establishing strong, supportive relationships with students [54]. They should also identify behaviors students find helpful, including listening, giving advice, following up to see whether the bullying has stopped, and beware of harmful behaviors such as exerting blame on victims or ignoring the issue [55].

For help-seeking barriers, the largest proportion of bullies closely followed by the uninvolved believed “I don’t think it is necessary” to seek help. Yet, underlying reasons for this choice seem to differ as most bullies and the uninvolved also chose “I don’t know where to seek help” and “I have no distressing issues/life difficulties”, respectively. Bullies’ unsureness of where to find support corresponds with them being the least likely group to seek help for seven of the eleven help-seeking sources. Major reasons may be the widespread punitive approach for dealing with bullying cases, as well as the general lack of awareness of bullies’ need for support. Teachers and counselors are less familiar with non-punitive strategies and generally support imposing sanctions for bullies [56], but recent studies indicate that zero-tolerance policies to punish bullies are ineffective, stigmatizing, and negatively affect the school climate [57,58]. This may explain why the majority of bullies express remorse but do not seek help or do anything after perpetrations [59]. A restorative justice approach may serve as a better alternative. Reintegrating bullies and victims back into the community through acknowledging and repairing the harm done, caring for others, and taking responsibility for their actions can improve youths’ use of adaptive shame management strategies and lower feelings of rejection by others after wrongdoing [60].

Hierarchical regressions found predominantly similar risk and protective factors predicting suicidal thinking and behaviors for all groups, failing to align with existing literature findings on each group having specific predictors. Risk factors that cross-cut the groups were psychological distress and diagnosis of a psychiatric disorder. For the involved and uninvolved groups, seeking virtual help and glorifying suicide were associated with heightened risks of suicidal thoughts and self-harm, respectively. Recurring factors shared among the groups suggest the robust nature of common risk factors in predicting suicidality. Psychiatric disorder and psychological distress have been widely validated across populations and countries as risk factors for suicide [13,24,61,62], and our study indicates it may be applicable not only in a general sense but also equally applicable for specific groups such as bullies, victims, and bully victims. Moreover, the Interpersonal Theory of Suicide provides further understanding concerning how the common risk factors of psychiatric disorder and psychological distress increases suicidal ideation and behaviors. Van Orden, Witte [63] explain that empirically validated risk factors are indicators of perceived burdensomeness, thwarted belongingness, or the acquired capability, and the simultaneous presence of these three constructs leads to the most dangerous form of suicidal desire. Psychiatric disorder and psychological distress can increase perceived burdensomeness as individuals may view themselves as expendable and unwanted or heighten self-beliefs as a liability for others, respectively.

Another notable finding was people involved with bullying who sought help from professionals, and online platforms were significantly more likely to exhibit suicidal thoughts and behaviors. Yet, those who sought help from their immediate circles were associated with lower suicide attempts. The contrasting findings may be due to difficulties in determining whether bullying or help-seeking occurs first. This corresponds with past conclusions on youth who report more frequent non-suicidal self-harm were more likely to seek help for this behavior [64], and interpreting bullied adolescents’ increased help-seeking as a successful result of encouragement from adults, anti-bullying interventions, and mental health professionals [65]. Help-seeking from different sources can be useful not only as a protective factor but also indicates that those who reach out for help require better understanding and support to cope with suicidal thoughts and behaviors. In addition to programs such as the Mental Health First Aid (MHFA), conversations about suicide courses that train individuals to converse with suicidal people [66], mental health professionals who currently undergo minimal training on suicide prevention [67,68], and services should also strive to develop higher levels of awareness and assistance towards those exhibiting signs of suicide. For instance, youth suicidality can be evaluated using cost and time-efficient suicide-screening assessments, including the Ask Suicide-Screening Questions (ASQ) Toolkit and the Suicide Behavior Questionnaire-Revised (SBQ-R) [69,70].

The findings of the present study should be considered in light of limitations. The self-report measure used may be prone to response biases, a possible confounding factor in which participants may give socially desirable responses for sensitive questions and select extreme or neutral responses as the primary question format rates on scales [71]. The sample may also not fully represent the general youth population in Hong Kong. Compared to Hong Kong’s youth population of slightly more males and 50.9% attaining post-secondary education, was mainly between the ages of 18–25, 68% males, and 79.69% had an education of post-secondary or above. This may be due to the survey being partly distributed through university channels. As participants were asked about their lifetime bullying experiences, it was difficult to account for specifics such as whether the participant transitioned from a bully to a victim or vice versa. The above limitations should therefore be weighed when understanding and interpreting the study results, and they should be addressed in future research.

Future research should also consider replicating our reported findings to fully examine the generalisability of the risk and protective factor profiles and predictors for adolescent bullies, victims, bully victims, and the uninvolved. Efforts can be dedicated towards developing interventions of building stronger peer support networks in school and work settings, lowering victim isolation through provisions of professional support, and encouraging bullies’ help-seeking behaviors through utilizing non-punitive approaches.

## 5. Conclusions

The current study aimed to identify the common and distinguishing risk and protective factors characterizing bullies, victims, bully victims, and those uninvolved. Based on the Hong Kong Online Survey on Youth Mental Health and Internet Usage, those involved as opposed to those uninvolved in bullying exhibited higher suicidal thoughts and behaviors. Bullies, victims, bully victims, and those uninvolved in bullying were found to possess distinct risk and protective factor profiles yet shared common predictors of suicidality. To further investigate the implications of the results, future research can continue to develop interventions suited to each group’s profiles, including peer support and inclusive measures to combat victim isolation and non-punitive counseling and school programs to reach out to bullies.

## Figures and Tables

**Table 1 ijerph-19-02828-t001:** Comparisons between the bully, victim, bully-victim, and uninvolved groups.

Variables	Bully (*n* = 119) ^a^	Victim (*n* = 413) ^b^	Bully Victim (*n* = 274) ^c^	Uninvolved (*n* = 1198) ^d^		
	*N* (%)/M (SD)	*N* (%)/M (SD)	*N* (%)/M (SD)	*N* (%)/M (SD)	X^2^/p/F	Phi (φ)/n^2^
Age	22.97 (4.63) ^a^	23.63 (5.19) ^a^	23.42 (5.16) ^a^	23.14 (5.17) ^a^	1.16	0.00
Gender					33.23 ***	0.13
Female	69 (57.98%) ^a^	300 (72.64%) ^b^	154 (56.2%) ^a,c^	855 (71.37%) ^b,d^		
Male	50 (42.02%) ^a^	113 (27.36%) ^b^	120 (43.8%) ^a,c^	343 (28.63%) ^b,d^		
Education					8.41	0.06
Post-secondary or above	98 (82.35%) ^a^	333 (80.63%) ^a^	211 (77.01%) ^a^	955 (79.72%) ^a^		
Secondary school Form 1-6	20 (16.81%) ^a^	74 (17.92%) ^a^	55 (20.07%) ^a^	201 (16.78%) ^a^		
Primary school	0 (0%) ^a^	1 (0.24%) ^a^	2 (0.73%) ^a^	10 (0.83%) ^a^		
Refuse to answer	1 (0.84%) ^a^	5 (1.21%) ^a^	6 (2.19%) ^a^	32 (2.67%) ^a^		
Occupation status					11.23	0.07
Full-time	30 (25.21%) ^a^	157 (38.01%) ^b^	107 (39.05%) ^a,b^	460 (38.4%) ^b^		
Part-time	42 (35.29%) ^a^	98 (23.73%) ^b^	73 (26.64%) ^a,b^	460 (38.4%) ^b^		
Not working	47 (39.5%) ^a^	158 (38.26%) ^b^	94 (34.31%) ^a,b^	460 (38.4%) ^b^		
Currently living with					12.44	0.08
Alone	7 (5.88%) ^a^	38 (9.2%) ^a^	21 (7.66%) ^a^	74 (6.18%) ^a^		
Family members	98 (82.35%) ^a^	324 (78.45%) ^a^	218 (79.56%) ^a^	1001 (83.56%) ^a^		
Friends	8 (6.72%) ^a^	32 (7.75%) ^a^	19 (6.93%) ^a^	88 (7.35%) ^a^		
Refuse to answer	6 (5.04%) ^a^	19 (4.6%) ^a^	16 (5.84%) ^a^	35 (2.92%) ^a^		
Family structure					23.00	0.11
Two-parent family	100 (84.03%) ^a^	330 (79.9%) ^a^	219 (79.93%) ^a^	966 (80.63%) ^a^		
Divorced parents	7 (5.88%) ^a^	49 (11.86%) ^a^	25 (9.12%) ^a^	146 (12.19%) ^a^		
Step-family	2 (1.68%) ^a^	3 (0.73%) ^a^	7 (2.55%) ^a^	8 (0.67%) ^a^		
One of the parents passed away	8 (6.72%) ^a^	21 (5.08%) ^a^	20 (7.3%) ^a^	65 (5.43%) ^a^		
Both parents passed away	0 (0%) ^a^	1 (0.24%) ^a^	1 (0.36%) ^a^	4 (0.33%) ^a^		
Other	2 (1.68%) ^a^	9 (2.18%) ^a^	2 (0.73%) ^a^	9 (0.75%) ^a^		
Source of Stress						
Academic	3.25 (1.06) ^a,b^	3.23 (1.22) ^a^	3.21 (1.18) ^a,b^	3.00 (1.19) ^b^	4.76 **	0.01
Job	2.94 (1.12) ^a,b^	3.06 (1.25) ^a^	2.93 (1.16) ^a,b^	2.74 (1.22) ^b^	6.13 ***	0.01
Financial circumstance	2.86 (1.37) ^a^	2.71 (1.26) ^a^	2.70 (1.27) ^a^	2.60 (1.23) ^a^	2.17	0.00
Social life	2.56 (1.07) ^a,b^	2.83 (1.14) ^a^	2.76 (1.19) ^a^	2.43 (1.10) ^b^	16.08 ***	0.02
Physical wellbeing	2.59 (1.15) ^a,b^	2.63 (1.16) ^a^	2.59 (1.19) ^a,b^	2.42 (1.10) ^b^	4.86 **	0.01
Mental wellbeing	2.91 (1.10) ^a,b^	3.15 (1.20) ^a^	3.05 (1.18) ^a^	2.71 (1.17) ^b^	17.78 ***	0.03
Relations with family	2.31 (1.12) ^a,b^	2.38 (1.17) ^a^	2.36 (1.15) ^a^	2.13 (1.08) ^b^	7.01 ***	0.01
Relations with partner	2.32 (1.23) ^a^	2.28 (1.23) ^a^	2.21 (1.24) ^a^	2.06 (1.16) ^a^	3.24 *	0.01
SOSS-SF						
SOSS-Glorification	2.16 (0.80) ^a,b^	2.37 (0.93) ^a^	2.37 (0.92) ^a^	2.15 (0.87) ^b^	9.53 ***	0.01
SOSS-Isolation	4.04 (0.78) ^a,b^	4.20 (0.73) ^a^	4.08 (0.85) ^a,b^	4.02 (0.86) ^b^	5.02 **	0.01
SOSS-Disgrace	2.14 (0.92) ^a^	1.98 (0.91) ^a^	2.07 (0.91) ^a^	2.05 (0.91) ^a^	1.19	0.00
SOSS-Selfishness	2.84 (1.11) ^a^	2.50 (1.15) ^b^	2.59 (1.17) ^a,b^	2.75 (1.16) ^a^	5.87 ***	0.01
SIDAS Score	4.85 (8.71) ^a,c^	8.16 (12.32) ^b^	6.85 (10.29) ^a^	2.93 (7.11) ^c^	41.25 ***	0.06
CHQ-12 Score	22.76 (6.33) ^a^	23.39 (6.46) ^a^	22.87 (6.48) ^a^	20.79 (6.18) ^b^	22.51 ***	0.03
Risk behaviors						
Considered Suicide	59 (49.58%) ^a^	259 (62.71%) ^b^	174 (63.50%) ^b^	416 (34.72%) ^c^	0.00 ***	0.27
Attempted suicide	12 (10.08%) ^a^	65 (15.74%) ^a^	34 (12.41%) ^a^	44 (3.67%) ^b^	0.00 ***	0.19
Injured self intentionally	33 (27.73%) ^a^	160 (38.74%) ^b^	129 (47.08%) ^c^	205 (17.11%) ^d^	0.00 ***	0.27
Social Withdrawal	16 (13.45%) ^a^	55 (13.32%) ^a^	38 (13.87%) ^a^	125 (10.43%) ^a^	0.19	0.05
Psychiatric disorders						
Major depressive disorder	6 (5.04%) ^a,c^	54 (13.08%) ^b^	31 (11.31%) ^a,b^	61 (5.09%) ^c^	0.00 ***	0.13
Schizophrenia	2 (1.68%) ^a,b^	13 (3.15%) ^a^	6 (2.19%) ^a,b^	9 (0.75%) ^b^	0.00 **	0.08
Social phobia	4 (3.36%) ^a,b^	14 (3.39%) ^a,b^	17 (6.20%) ^a^	19 (1.59%) ^b^	0.00 ***	0.10
Avoidant personality disorder	5 (4.20%) ^a^	9 (2.18%) ^a,b^	4 (1.46%) ^a,b^	9 (0.75%) ^b^	0.01 **	0.08
Help-Seeking Source						
FamilyMembers	31 (26.05%) ^a^	139 (33.66%) ^a^	83 (30.29%) ^a^	360 (30.05%) ^a^	0.37	0.04
Friends/Classmates/Colleagues	60 (50.42%) ^a,c^	246 (59.56%) ^a,b^	175 (63.87%) ^b^	618 (51.59%) ^c^	0.00 ***	0.10
Lover/LifePartner	37 (31.09%) ^a^	112 (27.12%) ^a^	67 (24.45%) ^a^	262 (21.87%) ^a^	0.04 *	0.07
Teachers/Tutors	8 (6.72%) ^a,b^	56 (13.56%) ^a^	25 (9.12%) ^a,b^	91 (7.60%) ^b^	0.00 **	0.08
Free Hotline Support	1 (0.84%) ^a^	14 (3.39%) ^a^	10 (3.65%) ^a^	21 (1.75%) ^a^	0.07	0.06
Medical Professionals	3 (2.52%) ^a,b^	29 (7.02%) ^a^	12 (4.38%) ^a,b^	39 (3.26%) ^b^	0.01 *	0.08
Social Workers/Counsellors	6 (5.04%) ^a^	66 (15.98%) ^b^	23 (8.39%) ^a^	89 (7.43%) ^a^	0.00 ***	0.12
Religious Services	7 (5.88%) ^a^	20 (4.84%) ^a^	11 (4.01%) ^a^	40 (3.34%) ^a^	0.30	0.04
Online friends (never met)	9 (7.56%) ^a,b^	38 (9.20%) ^a^	35 (12.77%) ^a^	44 (3.67%) ^b^	0.00 ***	0.14
Online social services	0 (0.00%) ^a,b^	9 (2.18%) ^a,b^	9 (3.28%) ^a^	10 (0.83%) ^b^	0.01 **	0.08
None of the above	37 (31.09%) ^a,b^	94 (22.76%) ^a^	67 (24.45%) ^a^	418 (34.89%) ^b^	0.00 ***	0.12
Help-Seeking Barriers						
Not sure how others will think of me	7 (5.88%) ^a^	15 (3.63%) ^a^	13 (4.74%) ^a^	32 (2.67%) ^a^	0.10	0.05
I don’t know where to seek help	5 (4.20%) ^a^	3 (0.73%) ^a^	5 (1.82%) ^a^	17 (1.42%) ^a^	0.06	0.06
I don’t want to bother others	8 (6.72%) ^a^	24 (5.81%) ^a^	22 (8.03%) ^a^	79 (6.59%) ^a^	0.71	0.03
I don’t think it is necessary	24 (20.17%) ^a,c^	53 (12.83%) ^a,b^	28 (10.22%) ^b^	224 (18.70%) ^c^	0.00 ***	0.09
I don’t think anyone can help me	8 (6.72%) ^a^	26 (6.30%) ^a^	23 (8.39%) ^a^	63 (5.26%) ^a^	0.23	0.05
I’m afraid it will leave a record and affect my future	5 (4.20%) ^a^	10 (2.42%) ^a^	6 (2.19%) ^a^	19 (1.59%) ^a^	0.19	0.05
I have no distressing issues/life difficulties	9 (7.56%) ^a^	36 (8.72%) ^a^	24 (8.76%) ^a^	215 (17.95%) ^b^	0.00 ***	0.13
Others	0 (0.00%) ^a^	5 (1.21%) ^a^	4 (1.46%) ^a^	12 (1.00%) ^a^	0.64	0.03

Different superscript letters (^a,b,c,d^) between the groups indicate significantly different group means. Significance level for letters: 0.05. * *p* < 0.05, ** *p* < 0.01, *** *p* < 0.001.

**Table 2 ijerph-19-02828-t002:** Means, standard deviations, and proportions for suicidal thoughts and behaviors in each group.

Variables	Bully (*n* = 80)	Victim (*n* = 309)	Bully Victim (*n* = 202)	Involved (*n* = 591)	Uninvolved (*n* = 755)
	M (SD)/ *N* (%)	M (SD)/ *N* (%)	M (SD)/ *N* (%)	M (SD)/ *N* (%)	M (SD)/ *N* (%)
SIDAS	5.13 (9.48)	8.57 (12.78)	6.71 (10.25)	7.47 (11.6)	3.18 (7.45)
Attempted suicide	8 (10%)	51 (16.51%)	25 (12.38%)	84 (14.21%)	29 (3.84%)
Injured self intentionally	22 (27.5%)	126 (40.78%)	97 (48.02%)	245 (41.46%)	133 (17.62%)

Individuals who did not seek any source of help and refused to answer for demographics were excluded from the regressions.

**Table 3 ijerph-19-02828-t003:** Summary of hierarchical multiple linear regression predicting SIDAS scores.

Variables	Bully (*n* = 80)	Victim (*n* = 309)	Bully Victim (*n* = 202)	Involved (*n* = 591)	Uninvolved (*n* = 755)
	Coefficient (β) (95% CI)				
Step 1: demographic variables					
Gender					
Female	Ref	Ref	Ref	Ref	Ref
Male					−0.37 (−1.48, 0.73)
Education level					
Below secondary school	Ref	Ref	Ref	Ref	Ref
Above secondary school	−2.63 (−8.02, 2.76)	−6.01 *** (−8.82, −3.19)	−1.99 (−4.91, 0.93)	−4.96 *** (−6.87, −3.05)	−1.99 ** (−3.19, −0.79)
Occupation					
Full-time (more than 30 h weekly)	Ref	Ref	Ref	Ref	Ref
Part-time (less than 30 h weekly)			2.87 (−0.03, 5.77)		
No work (did not seek work in the past 30 days)			1.99 (−0.80, 4.78)		
Family structure					
Non-nuclear family	Ref	Ref	Ref	Ref	Ref
Nuclear family		−0.38 (−3.07, 2.31)		−1.00 (−2.88, 0.89)	−0.92 (−2.15, 0.31)
Living with					
Live with others	Ref	Ref	Ref	Ref	Ref
Live alone					1.57 (−0.28, 3.43)
*R* ^2^	0.035	0.079	0.044	0.056	0.029
F	2.84	13.17 ***	3.06 *	17.50 ***	5.65 ***
AIC	589.02	2432.85	1513.48	4547.54	5163.47
Step 2: psychological and social factors					
Job					
Financial circumstance					
Social life					
Physical health	−1.84 (−7.34, 3.66)				
Mental well-being					
Relations with family/partner			0.87 (−2.13, 3.87)		
CHQ total	0.43 * (0.10, 0.75)	0.63 *** (0.45, 0.81)	0.58 *** (0.39, 0.78)	0.60 *** (0.47, 0.72)	0.40 *** (0.32, 0.48)
Social withdrawal		3.40 (−0.09, 6.89)		2.35 (−0.05, 4.75)	
Diagnosis of a psychiatric disorder	6.49 * (0.15, 12.83)	7.49 *** (4.24, 10.74)	5.76 *** (2.37, 9.14)	7.44 *** (5.21, 9.68)	4.91 *** (3.19, 6.64)
Schizophrenia					
Social anxiety					
Avoidant personality disorder					
*R* ^2^	0.241	0.383	0.301	0.338	0.211
Δ*R*^2^	0.206	0.304	0.257	0.282	0.182
ΔF	8.03 ***	56.93 ***	27.33 ***	93.58 ***	89.62 ***
AIC	575.86	2315.02	1456.42	4343.66	5010.93
Step 3: help seeking behaviors					
Help from immediate circle					
Help from professionals		3.38 ** (0.91, 5.84)		1.72 * (0.04, 3.40)	
Virtual help	10.49 ** (4.16, 16.82)	2.50 (−0.55, 5.55)	6.68 *** (3.82, 9.55)	4.46 *** (2.42, 6.49)	1.17 (−0.55, 2.89)
*R* ^2^	0.314	0.411	0.368	0.369	0.213
Δ*R*^2^	0.073	0.028	0.067	0.031	0.002
ΔF	8.60 **	7.88 ***	21.72 ***	15.42 ***	2.28
AIC	569.74	2304.64	1437.81	4319.32	5010.72
Step 4: stigma of suicide					
glorification		1.73 ** (0.50, 2.96)		1.28 ** (0.43, 2.14)	1.11 *** (0.56, 1.66)
isolation	1.84 (−0.55, 4.23)	1.49 (−0.04, 3.01)		1.02 * (0.05, 1.99)	
disgrace	−2.65 * (−4.75, -0.55)	−1.06 (−2.53, 0.41)		−0.84 (−1.82, 0.14)	−1.02 *** (−1.53, −0.50)
selfishness		−1.52 * (−2.70, −0.34)	−1.55 ** (−2.56, −0.53)	−1.34 ** (−2.15, −0.53)	
*R* ^2^	0.386	0.471	0.397	0.418	0.244
Δ*R*^2^	0.072	0.060	0.029	0.049	0.031
ΔF	4.20 *	8.42 ***	9.08 **	12.17 ***	15.34 ***
AIC	564.91	2279.44	1430.52	4279.63	4984.25

* *p* < 0.05, ** *p* < 0.01, *** *p* < 0.001.

**Table 4 ijerph-19-02828-t004:** Summary of hierarchical multiple logistic regression predicting suicide attempt.

Variables	Bully (*n* = 80)	Victim (*n* = 309)	Bully Victim (*n* = 202)	Involved (*n* = 591)	Uninvolved (*n* = 755)
	Odds Ratio (95% CI)				
Step 1: demographic variables					
Age		1.04 (0.96, 1.13)	1.06 (0.92, 1.20)		
Gender					
Female	Ref	Ref	Ref	Ref	Ref
Male	0.00 (0.00, Inf)				0.13 ** (0.02, 0.49)
Education level					
Below secondary school	Ref	Ref	Ref	Ref	Ref
Above secondary school	0.00 (0.00, Inf)	0.34 * (0.12, 0.95)		0.53 (0.28, 1.05)	0.31 ** (0.13, 0.76)
Occupation					
Full-time (more than 30 h weekly)	Ref	Ref	Ref	Ref	Ref
Part-time (less than 30 h weekly)			5.65 * (1.27, 29.17)		
No work (did not seek work in the past 30 days)			2.78 (0.56, 15.43)		
Family structure					
Non-nuclear family	Ref	Ref	Ref	Ref	Ref
Nuclear family			0.28 * (0.08, 0.98)		
Living with					
Live with others	Ref	Ref	Ref	Ref	Ref
Live alone				0.82 (0.43, 1.62)	
*R* ^2^	0.14	0.04	0.10	0.03	0.05
X^2^	5.36	7.05 *	10.67 *	8.34 *	11.38 **
Deviance	46.66	269.78	140.57	474.88	234.54
AIC	52.66	275.78	150.57	480.88	240.54
Step 2: psychological and social factors					
Academic/work stress		0.08 (0.00, 2.06)			
Financial circumstance	4.15758 × 10^40^ (0.00, Inf)		0.35 (0.09, 1.46)		
Social life		0.22 * (0.06, 0.76)		0.26 ** (0.11, 0.60)	
Physical health	0.00 (0.00, Inf)				
Mental well-being		9.88 (0.50, 1131.62)			6,305,976.44 (0.00, 1.447175 × 10^138^)
Relations with family/partner					
CHQ_total	474,157.22 (0.00, Inf)	1.08 * (1.02, 1.15)	1.14 ** (1.04, 1.25)	1.11 *** (1.06, 1.16)	1.07 * (1.00, 1.15)
Social withdrawal				6.73 *** (3.67, 12.52)	2.61 (0.71, 8.45)
Diagnosis of a psychiatric disorder	3.105252 × 10^114^ (0.00, Inf)	4.49 *** (1.99, 10.29)	11.43 *** (3.19, 44.81)	6.73 *** (3.67, 12.52)	4.33 ** (1.58, 11.60)
*R* ^2^	0.65	0.29	0.34	0.28	0.24
Δ*R*^2^	0.510	0.250	0.240	0.250	0.190
X^2^	24.24 ***	50.49 ***	29.72 ***	93.60 ***	40.10 ***
Deviance	22.42	219.30	110.85	381.28	194.44
AIC	36.42	235.30	126.85	393.28	
Step 3: help seeking behaviors					
Help from immediate circle	0.00 (0.00, Inf)	0.21 * (0.06, 0.73)	0.13 * (0.03, 0.64)	0.18 *** (0.07, 0.42)	
Help from professionals	0.00 (0.00, Inf)				2.28 (0.96, 5.44)
Virtual help	4.690596 × 10^40^ (0.00, Inf)	1.89 (0.76, 4.55)	3.36 * (1.06, 10.72)	3.04 *** (1.61, 5.69)	
*R* ^2^	1.00	0.31	0.45	0.35	0.26
Δ*R*^2^	0.350	0.020	0.110	0.070	0.020
X^2^	22.42 ***	5.51	13.75 **	25.30 ***	4.68 *
Deviance	0.00	213.79	97.10	355.98	189.76
AIC	20.00	233.79	117.10	371.98	205.76
Step 4: stigma of suicide					
glorification		1.51 (0.98, 2.35)			2.28 *** (1.45, 3.71)
isolation		2.50 ** (1.41, 4.69)		1.75 ** (1.18, 2.66)	
disgrace					
selfishness		0.48 *** (0.31, 0.72)		0.64 ** (0.49, 0.84)	
*R* ^2^	1.00	0.43	0.45	0.39	0.31
Δ*R*^2^	0.00	0.120	0.00	0.040	0.050
X^2^	0.00	26.75 ***	0.00	17.04 ***	13.00 ***
Deviance	0.00	187.04	97.10	338.94	176.76
AIC	20.00	213.04	117.10	358.94	194.76

* *p* < 0.05, ** *p* < 0.01, *** *p* < 0.001.

**Table 5 ijerph-19-02828-t005:** Summary of hierarchical multiple logistic regression predicting self-harm.

Variables	Bully (*n* = 80)	Victim (*n* = 309)	Bully Victim (*n* = 202)	Involved (*n* = 591)	Uninvolved (*n* = 755)
	Odds Ratio (95% CI)				
Step 1: demographic variables					
Age				1.03 (0.99, 1.07)	1.03 (0.99, 1.07)
Gender					
Female	Ref	Ref	Ref	Ref	Ref
Male	0.16 * (0.03, 0.63)	0.27 *** (0.13, 0.53)	1.07 * (1.01, 1.15)	0.68 (0.46, 1.01)	
Education level					
Below secondary school	Ref	Ref	Ref	Ref	Ref
Above secondary school	17.41 * (1.64, 496.86)	0.58 (0.29, 1.17)		0.87 (0.53, 1.44)	
Occupation					
Full-time (more than 30 h weekly)	Ref	Ref	Ref	Ref	Ref
Part-time (less than 30 h weekly)	0.09 ** (0.01, 0.44)				
No work (did not seek work in the past 30 days)	0.69 (0.12, 3.84)				
Family structure					
Non-nuclear family	Ref	Ref	Ref	Ref	Ref
Nuclear family		0.50 * (0.25, 1.00)		0.69 (0.44, 1.10)	
Living with					
Live with others	Ref	Ref	Ref	Ref	Ref
Live alone			0.33 (0.09, 1.07)	0.57 (0.28, 1.12)	
*R* ^2^	0.21	0.09	0.05	0.04	0.01
X^2^	12.37 *	22.41 ***	7.79 *	18.03 **	2.96
Deviance	81.74	395.38	271.92	783.92	699.97
AIC	91.74	403.38	277.92	795.92	703.97
Step 2: psychological and social factors					
Academic/work stress			0.20 (0.01, 1.42)		
Financial circumstance				0.64 (0.40, 1.02)	
Social life					1.61 (0.85, 3.29)
Physical health			0.43 * (0.19, 0.96)		
Mental wellbeing	106,367,913.95 (0.00, NA)	3.69 (0.94, 24.64)		2.31 (1.04, 5.67)	2.72 (1.03, 9.41)
Relations with family/partner			2.07 (0.93, 4.82)	1.63 (0.98, 2.78)	
CHQ_total		1.11 *** (1.05, 1.16)	1.07 * (1.02, 1.13)	1.06 *** (1.03, 1.10)	1.07 *** (1.03, 1.10)
Social withdrawal	0.00 (NA, 3.801751 × 10^74^)				1.61 (0.81, 3.09)
Diagnosis of a psychiatric disorder		1.77 (0.84, 3.81)	2.10 (0.84, 5.65)	2.36 ** (1.40, 4.03)	2.17 * (1.18, 3.92)
*R* ^2^	0.45	0.28	0.18	0.17	0.13
Δ*R*^2^	0.240	0.190	0.130	0.130	0.120
X^2^	17.12 ***	49.92 ***	21.55 ***	63.38 ***	61.24 ***
Deviance	64.62	345.46	250.37	720.54	638.73
AIC	80.62	359.46	266.37	742.54	652.73
Step 3: help seeking behaviors					
Help from immediate circle	14.73 * (1.69, 356.72)				
Help from professionals					
Virtual help		2.13 * (1.03, 4.54)		1.79 * (1.10, 2.94)	
*R* ^2^	0.45	0.30	0.18	0.19	0.13
Δ*R*^2^	0.00	0.020	0.000	0.020	0.000
X^2^	0.00	5.55 *	0.00	7.01 **	0.00
Deviance	64.62	339.91	250.37	713.52	638.73
AIC	80.62	355.91	266.37	737.52	652.73
Step 4: stigma of suicide					
glorification		1.33 (0.98, 1.81)		1.23 * (1.00, 1.50)	1.33 * (1.05, 1.69)
isolation					
disgrace	0.56 (0.26, 1.11)	0.73 (0.49, 1.08)		0.77 * (0.63, 0.95)	
selfishness		0.76 (0.56, 1.03)			0.79 * (0.66, 0.95)
*R* ^2^	0.48	0.36	0.18	0.21	0.17
Δ*R*^2^	0.030	0.060	0.000	0.020	0.040
X^2^	2.76	18.43 ***	0.00	10.90 **	15.33 ***
Deviance	61.85	321.48	250.37	702.62	623.40
AIC	79.85	343.48	266.37	730.62	641.40

* *p* < 0.05, ** *p* < 0.01, *** *p* < 0.001.

## Data Availability

All data and materials are available upon request.

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
