# Peer review of "Youth Bullying and Suicide: Risk and Protective Factor Profiles for Bullies, Victims, Bully-Victims and the Uninvolved"

_ijerph, 2022, doi:10.3390/ijerph19052828_

Round 1
Reviewer 1 Report
I want to thank the editor for the opportunity to review this work, which is, in my opinion, a highly relevant study on a subject that is of concern to both the scientific community and society. In addition, the article is well organised, and the ideas are clearly presented, which facilitates the understanding of the ideas presented in it. All this has allowed me to read it with great interest. As a result, I would like to make a few remarks to the authors.
The introduction sums up the state of the question quite well. However, perhaps the literature review on risk and protective factors could have been extended a little more, as well as including some consideration of the role of gender and age in previous works, given that these are variables included in the study presented.
In the methods section, I would suggest, although it is not a requirement in the journal's guidelines, to use the usual sequence of presentation of information. That is participants, measures, procedure and data analysis. In my opinion, this reorganisation would make it easier for the reader to find the information.
I think it would be necessary to specify some more data concerning the participants. For example, what is the participants' mean (and SD) age? The study has a very wide age range, and the lack of information makes it difficult to interpret the results. Similarly, a description of the gender distribution of the participants should be included here, as well as their current educational level (or occupation). In this sense, I would suggest including the description of the participants added within section 3.1.
Within the description of the measures used, I believe that the fit data of the instruments used with this sample should be provided, beyond Cronbach's Alpha. On the other hand, it is probably not necessary to divide the different instruments into different sub-sections.
Moreover, some additional clarification would be necessary for some of the measures. For example, in the instrument's case on sources of stress, some of the dimensions/items are aimed at the adult population. However, the sample includes participants as young as 11 years old. How have the responses of younger participants, who I understand have systematically left some of the dimensions unanswered (e.g. job-related), been handled?
It is also unclear to me the effect of asking about lifetime bullying experiences, especially when dealing with such a large sample in age. Could this fact be causing distortions in the data that are difficult to handle? For example, school bullying could be combined with workplace bullying experiences, of very different natures. Or it could be difficult to distinguish if we are dealing with a bully-victim profile or, on the contrary, with someone who changed role, from bully to a victim or vice versa, with years in between. On the other hand, and given that they are asked directly if they have participated in bullying episodes (as victims and aggressors), is a definition of the phenomenon provided so that participants can identify what kind of phenomenon is being referred to?
Concerning the results, the first paragraph that appears is from the template provided by the magazine. It should be removed. Regarding the organisation, I would suggest the authors organise the subsections according to the objectives, instead of the variables analysed in each case, as I understand it would facilitate the understanding of the analysis process that is being followed.
The discussion seems to me appropriate, taking up the main ideas that appear in work. However, I would suggest that the authors emphasise the protective role that some variables could have, not only their identification as aspects of risk.
Finally, in the references section, I suggest that the authors revise the format. The reference list is not following the ACS style. I would also suggest adding DOI when available (as indicated by the journal).
Reviewer 2 Report
This paper is an interesting account of a survey study of just over 2000 young people (predominantly students), exploring differences between bullies, victims, and bully-victims in relation to experiences of, and risk and protective factors for, suicide and related behaviours/thoughts.
The Introduction and Discussion sections are comprehensive, coherent and informative and I have very little to suggest in relation to these. I have made some comments, below, on the other sections, where I believe a little more detail would be beneficial:
Abstract - Lines 18-21 of abstract; it would be helpful to flag briefly whether/how the recommendations are derived specifically from the study findings.
Methods – a brief rationale for the choice of age range would be a nice addition.
Sources of stress; a little more information on this measure would be helpful. Were these items designed specifically for the study? Were all ratings summed to give an overall ‘stress’ score? The subtitle refers to ‘stress’, but the description of what was asked about refers to sources of ‘distress’ which is not quite the same. An example question wording would be useful here.
A justification for the decision to use the stigma of suicide scale would be helpful. The risk factors briefly mentioned as being of interest included ‘stigmatizing attitudes’ but the reference given for that is to a study that, I believe, is interested in stigmatizing attitudes towards mental health and how they can put a person at risk for suicide? I think a little more explanation on why stigma towards suicide is predicted to be a risk factor for suicide would be beneficial.
It would be helpful to indicate whether or not participants were given a specific definition for ‘deliberate self harm’ in the items asking about risk behaviours, or if they were to use their own interpretation (given the potential differences in interpretation of what this behaviour encompasses). Similarly, were participants given a definition of ‘bullying’ to guide their responses as to whether they had ever bullied/been bullied, or was this left to their own interpretation?
For ‘help seeking sources’, was there any specification in the question as to what they might have been seeking help with, or were they just asked whether they had ‘sought help (with anything at all) from…’?
Results
My main concern about the analysis is that there is no mention of correction for multiple comparisons; were any steps taken to guard against type 1 errors, given the very large number of comparisons made in the analysis? If not, how can you be confident that borderline significant findings (e.g. some p-values are very close to .05) are not false positives?
Lines 189-191 the guidance text has been left in the document.
A little more detail on the age range and distribution (lines 192-193) would be helpful when reporting the demographics. Edit – upon reading on, I see detail on this in Table 1. It would be helpful to signpost the reader to Table 1 in the text around this point.
In reporting the ANOVAs and post-hoc tests, it would be helpful to state the alpha value against which significance was judged (i.e. <0.05 or <0.01, etc?).
Some of your results of the tests of between groups differences are spelled out in the text; others are only reported briefly as omnibus test results without an account of the significance of post-hoc tests, and others are only reported as descriptives. At first I queried whether inferential stats had been run on these variables, but having reached Table 1 I see the information is presented there. I think it is important to signpost the reader to the table and flag that these tests have been run and the full results can be found there.
Your tables presenting the regression results need more information on what each column/number represents; specifically, I am guessing from context that the numbers in the main columns are (standardized?) Betas and confidence intervals but this information should be clearly indicated either in the column heading, legend, or elsewhere. I am also not sure why these are only presented for some columns, for certain variables.
Discussion
This is a really nice section where thoughtful explanations of the results are given, with reference to relevant prior literature.
Within the limitations it would be worth specifically mentioning that though the target age range was 11-35, the majority of participants were, I believe, in their early 20s?
